# *Terminalia chebula*-Assisted Silver Nanoparticles: Biological Potential, Synthesis, Characterization, and Ecotoxicity

**DOI:** 10.3390/biomedicines11051472

**Published:** 2023-05-18

**Authors:** Munusamy Tharani, Shanmugam Rajeshkumar, Khalid A. Al-Ghanim, Marcello Nicoletti, Nadezhda Sachivkina, Marimuthu Govindarajan

**Affiliations:** 1Nanobiomedicine Lab, Department of Pharmacology, Saveetha Dental College and Hospitals, SIMATS, Saveetha University, Chennai 600 077, Tamil Nadu, India; 2Department of Zoology, College of Science, King Saud University, Riyadh 11451, Saudi Arabia; 3Department of Environmental Biology, Sapienza University of Rome, 00185 Rome, Italy; 4Department of Microbiology V.S. Kiktenko, Institute of Medicine, Peoples Friendship University of Russia Named after Patrice Lumumba (RUDN University), Moscow 117198, Russia; 5Unit of Mycology and Parasitology, Department of Zoology, Annamalai University, Annamalainagar 608 002, Tamil Nadu, India; 6Unit of Natural Products and Nanotechnology, Department of Zoology, Government College for Women (Autonomous), Kumbakonam 612 001, Tamil Nadu, India

**Keywords:** *Terminalia chebula*, silver nanoparticles, mechanism, embryonic toxicology, antibacterial, antioxidant agent

## Abstract

In the current research, an aqueous extract of *Terminalia chebula* fruit was used to produce silver nanoparticles (Ag NPs) in a sustainable manner. UV-visible spectrophotometry, transmission electron microscopy (TEM), and scanning electron microscopy (SEM) were used to characterize the synthesized nanoparticles. Synthesized Ag NPs were detected since their greatest absorption peak was seen at 460 nm. The synthesized Ag NPs were spherical and had an average size of about 50 nm, with agglomerated structures, as shown via SEM and TEM analyses. The biological activities of the synthesized Ag NPs were evaluated in terms of their antibacterial and antioxidant properties, as well as protein leakage and time-kill kinetics assays. The results suggest that the green synthesized Ag NPs possess significant antibacterial and antioxidant activities, making them a promising candidate for therapeutic applications. Furthermore, the study also evaluated the potential toxicological effects of the Ag NPs using zebrafish embryos as a model organism. The findings indicate that the synthesized Ag NPs did not induce any significant toxic effects on zebrafish embryos, further supporting their potential as therapeutic agents. In conclusion, the environmentally friendly production of Ag NPs using the extract from *T. chebula* is a promising strategy for discovering novel therapeutic agents with prospective uses in biomedicine.

## 1. Introduction

The distinct characteristics of nanomaterials have garnered increased interest from scientists, thereby facilitating diverse biomedical implementations [1]. The fabrication of nanoparticles with diverse structures can be accomplished through physical, biological, and chemical methodologies. However, physical and chemical methods often result in low yields, high costs, high energy consumption, and environmental damage. Biological synthesis, which employs plants and microbes as reducing agents, is one of the three synthesis approaches [2,3]. Because of their superior thermal, strong optoelectronic, catalytic, and surface volume ratio properties, metal nanoparticles play an essential role in the healthcare sectors. This is due to the fact that they have great physicochemical features. Silver, gold, and platinum are characterized as the “noble metal nanoparticles” among all of the other metal nanoparticle types [4,5]. Silver nanoparticles have been paid special attention for their exclusive property, i.e., targeting highly resistant microbes via their antimicrobial mechanism. Silver nanoparticles have tremendous antimicrobial properties and they have been one of the most explored noble nanoparticles to date [6]. Subsequently, further research must be conducted to address the disadvantages of ordinary antimicrobial formulations and combat multidrug-resistant pathogens [7,8].

Silver nanoparticles synthesized via chemical routes often produce toxic, non-eco-friendly products. Recently, greener routes, such as plants, have been chosen as an alternative platform to synthesize silver nanoparticles safely [9,10]. Synthesized nanoparticles derived from plants have attracted interest because of their potential benefits to human and animal health. Oves et al. [11] studied silver nanoparticles synthesized from *Conocarpus lancifolius*, which demonstrated excellent antibacterial activity against *Streptococcus pneumoniae* and *Staphylococcus aureus*, as well as antifungal activity against *Rhizopus stolonifer* and *Aspergillus flavus*. Similarly, Selvaraju et al. [12] synthesized silver nanoparticles using *Ethanolic propolis* extract and coated them over plain surgical sutures. The propolis-extract-coated sutures were tested for antimicrobial activity and showed significant effects against clinical infectious pathogens such as *Aspergillus flavus*, *Escherichia coli,* and *Staphylococcus aureus*.

*Terminalia chebula* dried fruit is a member of the Combretaceae family. It is known by various common names such as dark myrobalan, Haritaki (Sanskrit and Bengali), Kadukkai (Tamil), and Harad (Hindi). In Ayurvedic remedies, *T. chebula* is commonly referred to as the “King of Medicine” and is used to treat various human complications due to its rich phytoconstituent composition, which includes flavonoids, tannins, sterols, amino acids, and resins. Gallic acid, Chebulagic acid, corilagin, and chebulic acid are some of the tannins responsible for the various applications of *T. chebula*, including its anti-inflammatory, antioxidant, antidiabetic, antimicrobial, anticarcinogenic, and anti-aging properties [13,14]. The current research work focuses on investigating the antibacterial mechanism, antioxidant potential, and toxicology evaluation of green synthesized AgNPs, with the objective of developing them as valuable therapeutics for future drug discovery.

## 2. Materials and Methods

### 2.1. Materials

*T. chebula* was procured from a botanical dispensary located in Poonamallee, India, while Mueller-Hinton agar and broth were procured from Hi-media, Mumbai, India. The chemicals DPPH and Bradford reagent were purchased from Sigma Aldrich, India. Furthermore, zebrafish (*Danio rerio*) were acquired from a supplier based in Kolathur, Chennai, India.

### 2.2. Preparation of Plant Extract

The preparation of *T. chebula* extract involved the dissolution of one gram of powdered dried fruit in 100 mL of distilled water. The solution was subjected to boiling for a duration of 15–20 min using a heating mantle set at a temperature of 70 °C. The filtration process was carried out using Whatman filter paper. The extract obtained from the filter was kept under refrigeration until the production of nanoparticles. 

### 2.3. Synthesis of T. chebula-Mediated Silver Nanoparticles: TCF-AgNPs

A solution of 1 mM AgNO_3_ in 90 mL distilled water was used to synthesize the silver nanoparticles. This solution received 10 mL of filtered *T. chebula* extract. Then, it was stirred at 750 rpm for 24 h using a magnetic stirrer. UV-visible measurements tracked the response at 24 h intervals. The pellet was obtained through centrifugation at a speed of 8000 revolutions per minute for duration of 10 min. The silver nanoparticles were subjected to a purification process whereby the supernatant was eliminated and the pellet was subjected to two rounds of washing using double-distilled water and ethanol. The pellet was stored within a hermetically sealed Eppendorf tube for the purposes of both characterization and subsequent utilization. 

### 2.4. Characterization of Green Synthesized AgNPs

The absorption bands of the synthesized TCF-AgNPs were determined using a double-beam ultra violet-visible spectrophotometer (ESICO—model 3375) with a wavelength range of 350 nm to 550 nm. The morphological features were examined through the utilization of scanning electron microscopy (SEM) (JEOL FE SEM IT-800) and transmission electron microscopy (TEM). The identification of functional groups was accomplished through the utilization of Fourier transform infrared spectroscopy (FT-IR) technology, specifically the BRUKER model.

### 2.5. Biomedical Applications

#### Antioxidant Activity—DPPH Assay

The 1,1-diphenyl-2-picrylhydrazyl (DPPH) test was used in order to evaluate the level of antioxidant activity shown by the TCF-AgNPs. *T. chebula* dried fruit extract served as the mediator in the production of silver nanoparticles, which were made in a range of concentrations from 100 to 1000 µg/mL. In order to carry out the experiment, the nanoparticles were mixed with one mL of methanol containing 0.1 mM of DPPH and 450 mL of a buffer containing 50 mM of TRIS-HCL with a pH of 7.4. After that, the resultant combination was left to ferment for a period of half an hour. To measure the decrease in DPPH free radicals, the absorbance at 517 nm was continuously monitored. Butylated Hydroxy Toluene and ascorbic acid were used as standards to compare the results.

To calculate the percentage of inhibition, the following equation was used:% Inhibition = (Absorbance of control − Absorbance of sample)/(Absorbance of control) × 100

### 2.6. Protein Leakage Analysis or Bradford Assay 

The Bradford test was used in order to carry out the protein leakage investigation. Bacterial cells from *Staphylococcus aureus*, *Pseudomonas aeruginosa*, and *Escherichia coli* were subjected to various concentrations of green produced AgNPs over a period of 24–48 h. These concentrations were 100 µg, 250 µg, 500 µg, 750 µg, and 1000 µg, respectively. Following treatment, the bacterial suspension underwent centrifugation at 6000 rpm for 15 min, separating a supernatant phase, which was subsequently collected. A total of 200 µL of supernatant was used for each sample and added to 96-well ELISA plates. To these, 800 µL of Bradford reagent was added and maintained for 10 min of incubation in a dark environment. Amoxicillin acted as the standard. The sample’s optical density (OD) was measured at 595 nm.

### 2.7. Time-Kill Curve Assay

A time-kill curve assay was conducted to assess the bactericidal properties and concentration-dependent relationship between *T. chebula*-mediated silver nanoparticles and the net growth rate of *Pseudomonas aeruginosa, Escherichia coli*, and *Staphylococcus aureus* over regular time intervals. The assay involved culturing the three wound pathogens in Mueller Hinton Broth supplemented with varying concentrations of silver nanoparticles (100 µg, 250 µg, 500 µg, 750 µg, and 1000 µg), followed by time-kill curve analysis. After a pre-incubation period of four hours in a medium devoid of any antimicrobial agents, growth curves were carried out before the test to ensure that all pathogens had reached a stable early-to-mid log phase. An inoculum consisting of 0.5 McFarland of each pathogen was created in sterile phosphate-buffered saline. This inoculum was collected from cultures that had been cultivated on Mueller Hinton agar plates at 37 °C for 18–20 h. After that, 30 µL of the inoculum was diluted in 15 mL of antimicrobial-free Mueller Hinton Broth medium that had been pre-heated to 37 °C, and 90 µL of the resultant mixture was distributed evenly over each well of a 96-well ELISA plate. To each well containing 90 µL of pre-incubated wound pathogens, 10 µL of *T. chebula*-mediated silver nanoparticles at five different concentrations was added, along with the untreated control.

### 2.8. Antibacterial Activity

The agar well diffusion technique, as previously described by Liaqat et al. [15], was employed to evaluate the antibacterial effectiveness of TCF-AgNPs. The antimicrobial efficacy of silver nanoparticles was evaluated against three types of wound pathogens, namely *Pseudomonas aeruginosa*, *Escherichia coli*, and *Staphylococcus aureus*. For this test, Mueller Hinton agar was selected and utilized to measure the zone of inhibition. To prepare the Mueller Hinton agar, it was sterilized at 121 °C for 15 min at 15 lbs and then allowed to solidify. Four sterile 9 mm polystyrene tips were used to create wells on the surface of the agar plates. Each wound pathogen was swabbed onto the surface of its respective Mueller Hinton agar plate using a sterile cotton swab. TCF-AgNPs were applied to the wells at five different concentrations to test their antibacterial effectiveness. Subsequently, the plates were subjected to incubation at a temperature of 37 °C for a duration of 24 h. Following the incubation period, a meticulous examination of the plates was conducted to observe the zone of inhibition encircling the wells, and subsequently, the percentage of growth inhibition was computed.

### 2.9. Zebrafish Embryonic Toxicology Evaluation of T. chebula-Mediated Silver Nanoparticles

#### 2.9.1. Fish Maintenance and AgNP Exposure

The *Danio rerio*, commonly known as the wild-type zebrafish, were procured from local vendors in India and were housed in distinct tanks under regulated environmental conditions. The experimental conditions encompassed specific parameters, namely a temperature of 28.2 °C, a light/dark cycle of 14:10 h, and a pH range spanning from 6.8 to 8.5. The zebrafish were given dried blood worms or the ideal meal twice a day. Both of these foods are commercially accessible. In order to acquire zebrafish embryos, one female and three male zebrafish were put in a breeding tank together. When this had been performed, viable eggs were produced, which were then retrieved and thoroughly washed at least three times using newly made E3 media that did not contain methylene blue. Fertilized eggs were plated out in 6-well, 12-well, and 24-well culture plates, with 20 embryos in 2 mL of solution per well. There were three sets of experimental treatment and control groups. The E3 medium used in the experiment was dosed with a newly prepared stock solution of TCF-AgNPs at one of five distinct concentrations. The solution was sonicated for 15 min to disperse the nanoparticles while maintaining a pH range of 7.2–7.3. Healthy fertilized embryos were subjected to various concentrations of AgNPs ranging from 0 to 1000 µg/L for 24 to 96 h post fertilization. The E3 medium was supplemented with AgNPs and subsequently utilized for the incubation of the embryos. Control groups were also included in the experiment. Dead embryos were removed from the nanoparticle-exposed groups every twelve hours. To prevent light interference, all experimental plates were covered in foil and maintained at a temperature of 28 °C.

#### 2.9.2. Zebrafish Embryo Evaluation

The developmental stages of zebrafish embryos were meticulously observed during the exposure period following fertilization using a stereo microscope. During a period of 24 to 78 h post-fertilization, the zebrafish embryos were subjected to varying concentrations of silver nanoparticles, ranging from 0 to 1000 µg/L, including 100, 250, 500, 750, and 1000 µg/L. The study endpoints consisted of embryonic mortality and hatching rates evaluated at 24-h intervals. Malformations observed among the embryos and larvae in both the control and treatment groups were identified and documented using a COSLAB—model: HL-10A light microscope. Photographs of any malformed embryos were taken, and the percentage of abnormal embryos was documented every 24 h.

### 2.10. Statistical Analysis

The experiments were subjected to a replication process of three iterations, and the outcomes are presented in the form of means and standard errors. Graphpad 9.4.1 was used for the statistical analysis. The study employed a two-way analysis of variance (ANOVA) followed by a Bonferroni post hoc test to ascertain the presence of statistically significant differences between the control drug and the nanomaterial samples. The data obtained from the time-kill curve investigation were subjected to statistical analysis using one-way analysis of variance (ANOVA) and Dunnett’s post hoc test. Different concentrations of the standard drugs and the test samples were evaluated using a *p* ≤ 0.05 significance threshold.

## 3. Results and Discussion

### 3.1. Visual Observation

The process of green synthesis was employed to produce silver nanoparticles (TCF-AgNPs) by utilizing *T. chebula* dried fruit extract as a reducing and stabilizing agent. This method involves the use of biological materials, such as plants, to facilitate the synthesis process. As demonstrated in Figure 1, the phytochemicals in *T. chebula* dried fruit extract have a reducing effect, as shown by the color shift from light yellow to dark greyish-brown. Synthesized silver nanoparticles (AgNPs) take on a specific color throughout their formation and nucleation processes, making color a crucial aspect. According to the study, a conspicuous alteration in hue from light yellow to brownish-yellow is a robust indication of the swift generation and nucleation of AgNPs. The manifestation of the brown hue is attributed to the surface plasmon resonance phenomenon and the conversion of silver ions (Ag^+^) to elemental silver (Ag^0^) in aqueous extracts. Furthermore, the visual observation of a colour change in the reaction medium containing AgNPs was indicative of the reduction of metal ions, as reported in [16]. The alteration of colour is a significant indicator in the synthesis of silver nanoparticles, and it has the potential to serve as a means of monitoring the advancement of the reaction. The visual observation approach was followed up with ultraviolet-visible spectral analysis to further check its accuracy.

### 3.2. UV-Visible Spectroscopy

The UV-visible absorption spectra of AgNPs synthesized using *T. chebula* dried fruit extract after 24 h are depicted in Figure 2. UV-visible spectra were obtained at specific time intervals (1 h, 8 h, and 24 h) for TCF-AgNPs. The surface plasmon resonance (SPR) bands of the TCF-AgNPs were observed at different wavelengths for each time interval. After a synthesis time of 1 h, the silver nanoparticles showed an SPR band with a maximum wavelength of 375 nm. This band is indicative of the nanoparticles’ ability to absorb Ag^+^ ions. After a period of 24 h, the silver nanoparticles exhibited a maximum absorption peak at a wavelength of 460 nm. At a maximum wavelength of 460 nm, this unique absorption peak indicates the synthesis of green-synthesized TCF-AgNPs and validates the reduction process that occurs when Ag^+^ is converted to Ag^0^. The absorbance of the silver nanoparticles increased steadily with time, indicating the continuous production of more AgNPs until the reaction reached completion. The absorbance of the nanoparticles was evaluated using a double-beam UV-visible spectrophotometer, and after 24 h, there was no change, indicating that the reduction of the silver nanoparticles was complete. Dissolving silver nanoparticles in water caused their surface plasmon resonance to become excited, altering the color of the nanoparticles from yellow to greyish brown [17]. The utilization of UV-Vis spectroscopy is a crucial method for confirming the establishment and endurance of AgNPs within aqueous solutions [18]. The process of AgNP growth can be monitored via UV-Vis spectroscopy, wherein the characteristic absorption maxima of surface plasmons ranging from 391 to 453 nm are exhibited [19]. Recently, a study has reported the maximum absorption peak of *Gleichenia-pectinate*-extract-mediated silver nanoparticles to be 460 nm, which correlates with the current study looking at *T. chebula*-dried-fruit-extract-mediated silver nanoparticles [20]. UV-Vis spectroscopy is a valuable analytical technique that can be employed to confirm the synthesis of AgNPs and investigate their temporal evolution.

### 3.3. SEM and TEM Analysis

The surface morphology of AgNPs synthesized by *T. chebula* was analyzed using SEM. TCF-AgNPs, as shown in Figure 3a, have a spherical aggregate form. Transmission electron microscopy (TEM) was used to analyze the silver nanoparticles for their size, shape, and overall morphology. TCF-AgNPs were polydisperse and spherical, with an average size of around 50 nm, as shown in the TEM picture (Figure 3b), which confirms their successful synthesis. Notably, the small size of TCF-AgNPs is an important characteristic that contributes to their potent antimicrobial activity [21,22]. Previous studies have used TEM micrographs to ascertain the size and form of the silver nanoparticles mediated by *Daucus carota*. Particle size distribution micrographs showed an average of 18.26 nm [23]; the nanoparticles were found to be spherical and in the face-centered cubic phase. Silver nanoparticles mediated by *Tectona grandis* seed extract were spherical and around 10–30 nm in size, according to TEM imaging in another work [24]. *The extract of Syzygium cumini* has the potential to function as a reducing agent during the synthesis of silver nanoparticles. According to transmission electron microscopy studies, the synthesized nanoparticles exhibit a spherical shape with an average size ranging from 10 to 100 nm [25]. In one work, silver nanoparticles were synthesized using an extract from the *Origanum vulgare* L. plant, and their average size was reported using transmission electron microscopy examination. The findings revealed two distinct average sizes, with the first measuring 38 ± 10 nm and the second measuring 7 ± 3 nm [26]. 

### 3.4. FT-IR

The Fourier Transform Infrared (FTIR) analysis conducted on the silver nanoparticles synthesized with *T. chebula* extract revealed the presence of distinct peaks at different wavenumbers, as depicted in Figure 4. The presence of O-H groups is indicated by a broad peak observed at 3363.43 cm^−1^, which suggests that the hydroxyl groups present in the extract may have functioned as a reducing agent in the synthesis of silver nanoparticles. The spectral feature observed at 1605.81 cm^−1^ can be ascribed to the stretching vibrations of C=C bonds within aromatic rings. This characteristic peak is likely to have emanated from the phenolic compounds that were detected in the extract. The observation of a peak at 578.79 cm^−1^ provides evidence for the existence of Ag-O bonds, which suggests the creation of silver nanoparticles. Other peaks observed at 1702.15, 1443.49, 1317.36, 1182.59, 1052.08, 869.61, and 754.46 cm^−1^ suggest that the presence of various functional groups could be from the extract or biomolecules adsorbed onto the surface of the nanoparticles. The FTIR study shows that *T. chebula* extract may have helped the silver nanoparticles to be more biocompatible and active by acting as a reducing and capping agent during their production. The silver nanoparticles mediated by *T. chebula* displayed two strong bands in FTIR analysis, one at 3438 cm^−1^, which corresponds to O-H stretching and H-bonding, and another at 708 cm^−1^, corresponding to C=C bending, as shown in previous work. The research indicates that silver nanoparticles may be reduced and capped due to the presence of phytoconstituents [27]. The FTIR examination of silver nanoparticles synthesized from *T. chebula* seed extract was published in another work [28]; however, the particular peaks identified were not specified. Previous studies found that silver nanoparticles mediated by *Aspergillus niger* exhibited unique peaks at 548.38 cm^−1^, 1636.17 cm^−1^, and 3347.85 cm^−1^ [29].

### 3.5. Antioxidant Activity

#### DPPH Radical Scavenging Potential

Antioxidants are chemical compounds that neutralize free radicals by donating an electron or hydrogen atom, thereby preventing oxidative damage to cells and tissues. The capacity of antioxidants to convert the stable radical DPPH to a non-radical state was measured in this work using the DPPH test, and the newly synthesized TCF-AgNPs were shown to have high antioxidant activity. DPPH is a stable radical with peak absorption at 520 nm [30]. TCF-AgNPs were utilized in this research at doses of 100, 250, 500, 750, and 1000 g/mL. As seen in Figure 5, the concentration-dependent shift in color from violet to light yellow resulted from the conversion of DPPH from its radical to non-radical state [31].

A DPPH radical assay was performed along with ascorbic acid as a standard. In this study, the DPPH test results exhibited these silver nanoparticles as good free radical scavengers, which are depicted in Figure 6. The 100 and 1000 µg/L concentrations of ascorbic acid showed scavenging activity of about 45.5% and 87.2%, respectively. The statistical analysis of the data using the two-way ANOVA test revealed that the empirical p-value was below 0.05. This finding suggests that the TCF-AgNPs resulted in a noteworthy augmentation of antioxidant activity at 500, 750, and 1000 µg/mL concentrations, compared to ascorbic acid. At a concentration of 500 µg/mL, the TCF-AgNPs revealed a percentage inhibition of 93.47 ± 8.19%, which was equivalent to that of the standard drug (109.81 ± 5.78%). At a concentration of 1000 µg/mL, the TCF-AgNPs showed a substantial (*p* < 0.001) enhancement in their antioxidant activity compared to the other concentrations tested (124.33 ± 17.8%), which was also equivalent to the standard drug (137.29 ± 5.79%). Other concentrations (100, 250 µg/mL) showed significant differences (*p* < 0.05) between the TCF-AgNPs and the standard drug. The IC_50_ values of the TCF-AgNPs and standard drug (ascorbic acid), which represent the concentrations needed for a 50% reduction of the DPPH radical, were determined to be 55.67 ± 11.1 and 65.87 ± 0.03 µg/mL, respectively. The *T. chebula*-mediated AgNPs demonstrated potent free radical scavenging ability against DPPH free radicals, generated as a result of oxidative stress. Previous research has reported the synthesis of Medicago-sativa-extract-mediated silver nanoparticles and their ability to scavenge free radicals to about 78%. This finding highlights the capping activity of secondary metabolites present in the plant extract, which results in higher antioxidant activity [32]. In another study, *Erythrina suberosa*-leaf-extract-mediated silver nanoparticles were reported to show enhanced antioxidant activity at IC_50_ 30.04 µg/mL [33].

### 3.6. Antibacterial Activity

Pathogenic Gram-positive and Gram-negative bacteria such *Escherichia coli*, *Pseudomonas aeruginosa*, and *Staphylococcus aureus* were tested for susceptibility to silver nanoparticles engineered using eco-friendly methods. Using a two-way ANOVA test, we discovered that the experimental *p*-value was less than 0.05, signifying that the antibacterial activity at 500, 750, and 1000 µg/mL was significantly higher than that of the standard antibiotic amoxicillin (Figure 7).

Notably, TCF-AgNPs exhibited significantly higher antibacterial potential against *P. aeruginosa* at these concentrations, as evidenced by larger zone of inhibition sizes of 34.00 ± 1.15 mm, 35.00 ± 1.15 mm, and 37.00 ± 1.15 mm, compared to the standard drug’s values of 32.00 ± 1.15 mm, 33.33 ± 0.88 mm, and 36 ± 1.15 mm (*p* < 0.001). Conversely, *S. aureus* and *E. coli* exhibited comparable zone of inhibition sizes at these concentrations.

Elevating the concentration of silver nanoparticles for treating pathogenic bacteria resulted in an increased zone of inhibition, with a more pronounced effect observed in Gram-negative than Gram-positive bacteria. The mechanism underlying the antimicrobial activity of silver ions remains incompletely understood; however, it is suggested that they penetrate the microbial cell and cause DNA condensation, ultimately leading to cell death. As silver nanoparticles belong to the heavy metal category, they bind with microbial proteins and inactivate them by attaching them to the thiol group. The antibacterial efficacy of silver nanoparticles is also influenced by their size, with smaller particles exhibiting their larger surface area and capability resulting in higher activity to attach to the bacterial cell wall, leading to the disturbance of the permeation and respiration functions of the cell [34,35,36].

### 3.7. Protein Leakage Analysis

This study has explored the impact of silver nanoparticles on three wound pathogens: *Pseudomonas aeruginosa, Staphylococcus aureus*, and *Escherichia coli*. Our findings demonstrate that elevating the concentration of TCF-AgNPs leads to higher protein leakage from bacterial cells compared to the standard antibiotic amoxicillin, as shown in Figure 8. Moreover, significant differences were observed between Gram-positive (*S. aureus*) and Gram-negative (*P. aeruginosa* and *E. coli*) strains, with Gram-negative strains exhibiting a greater increase in protein concentration. Our results indicate that biosynthesized AgNPs exert potent effects by inducing oxidative stress on bacterial cells, disrupting protein content. The negative charge of AgNPs is responsible for damaging the bacterial cell wall membrane, which results in cell lysis. It was noted that the concentration of *P. aeruginosa* showed a direct correlation with the amount of cellular protein of AgNPs synthesized using TCF released, suggesting that TCF-AgNPs can be used as effective antimicrobial agents [37,38].

### 3.8. Time-Kill Curve Assay

In this study, a time-kill kinetics study was performed to investigate the antimicrobial activity of TCF-AgNPs against three wound pathogens, *Pseudomonas aeruginosa, Staphylococcus aureus*, and *Escherichia coli*, at various concentrations ranging from zero to 1000 µg/L. The decrease in colony-forming units, or bactericidal activity, was used to determine the efficacy of TCF-AgNPs in killing the bacteria. The time-kill curve assay allowed for the monitoring of the antimicrobial efficacy of TCF-AgNPs over time, in relation to the growth stages of the bacteria, such as the lag, exponential, and stationary phases. Our results indicate that the TCF-AgNPs induced a rapid and enhanced bactericidal effect against *P. aeruginosa* at concentrations equal to or above the minimum inhibitory concentration (MIC).

The time-kill kinetics profile of TCF-AgNPs against the test organisms revealed a decrease in the count of viable cells over the first 1–2 h for *P. aeruginosa* and *S. aureus* followed by a gradual rise up to the 4th hour for *E. coli*, compared to the control (Figure 9a). We found that the course of antimicrobial action was concentration-dependent and bacteriostatic for TCF-AgNPs. Moreover, the TCF-AgNP area under the curve (AUC) against *P. aeruginosa* (*p* < 0.001), *S. aureus* (*p* < 0.05), and *E. coli* (*p* < 0.05) at high concentrations exhibited a reduction in bacterial cells when compared to the control group (Figure 9b). The unique size of the synthesized TCF-AgNPs enhanced their ability to penetrate deeply into the bacterial cells, causing cell damage and resulting in a rapid and enhanced bactericidal effect against *P. aeruginosa*. The bacteriostatic effect and concentration dependence of TCF-AgNPs also indicate their potential as effective antimicrobial agents against other wound pathogens [39,40]. Prior research has employed time-kill curve assays to assess the antimicrobial efficacy of silver nanoparticles against diverse bacterial strains. The impact of silver nanoparticles on the growth of *S. aureus* and *P. aeruginosa* was examined by Gurunathan et al. [41] through the use of time-kill curves. The results revealed that silver nanoparticles exhibited a bactericidal effect against the strains, significantly reducing colony-forming units (CFUs) over time. Fayaz and colleagues [42] employed time-kill curves to evaluate the antimicrobial efficacy of silver nanoparticles against *S. aureus* and *E. coli*. The findings demonstrated that silver nanoparticles had a concentration-dependent bactericidal effect against the strains, with complete eradication of the bacterial population at higher concentrations.

### 3.9. Zebrafish Study

#### Hatching and Viability Rate

Various concentrations of TCF-AgNPs (100–1000 µg/mL) were used to treat the zebrafish embryos before 4 h post fertilization (hpf). The treated embryos showed developmental abnormalities, which were noted. An increasing concentration of TCF-AgNPs gradually affects the hatching rate of the embryos (Figure 10a). The hatching rate of the embryos at 100–500 µg/mL concentration was 94.92 ± 0.00%, while the 750 µg/mL concentration of TCF-AgNPs was 76.67 ± 3.9%, and for 1000 µg/mL it was 70 ± 5.3%. The concentrations of 100, 250, and 500 exhibited a significant value of less than (*p* < 0.01), were found to be moderately significant (*p* < 0.05), and caused delayed hatching at high concentrations of nanoparticles ranging from 750 to 1000 µg/mL.

The study evaluated the TCF-AgNP viability rate from 0 hpf to 96 hpf. The results demonstrated a consistent viability rate across all concentrations (100–1000 µg/mL) tested, indicating no toxicity when compared to the control group. However, concentrations of 750 and 1000 µg/mL TCF-AgNPs were found to have a (*p* < 0.0001) highly significant effect on the mortality rate, with both concentrations causing 80 and 100% mortality, respectively, by 96 hpf. The results suggest that while lower concentrations of TCF-AgNPs do not exhibit significant toxicity, higher concentrations can be highly detrimental to zebrafish embryo survival.

The study revealed that the exposure of zebrafish embryos to TCF-AgNPs at concentrations ranging from 100 to 500 µg/mL did not significantly affect viability rates from 0 hpf to 96 hpf (Figure 10b). However, concentrations of 750 and 1000 µg/mL caused highly significant effects on mortality, resulting in 80 and 100% mortality at 96 hpf, respectively. The lethality concentration of silver nanoparticles was determined to be 750 µg/mL, and no developmental abnormalities were observed at any concentration (Figure 11). These findings indicate that the synthesized TCF-AgNPs were generally less toxic to zebrafish embryos.

Machado et al. [43] conducted a study that demonstrated the potential of utilising green synthesis techniques to produce nanoparticles with decreased toxicity. *Cystoseira macroalgae* was extracted to synthesize gold nanoparticles, demonstrating their non-toxic nature even at high concentrations of 1.25 and 2.5 mM. The importance of this research stems from the utilisation of conventional chemical and physical synthesis techniques that involve the use of perilous reagents. This practise can elevate the toxicity of nanoparticles and reduce their biocompatibility. Furthermore, using eco-friendly materials in green synthesis makes them more sustainable and less environmentally harmful. Therefore, this study suggests that green synthesis methods could be a promising approach for producing nanoparticles with reduced toxicity and improved biocompatibility.

The impact of silver nanoparticles on the embryonic development of zebrafish was investigated by Xu et al. [44]. Different concentrations and incubation times of silver nanoparticles (AgNPs@PVP and AgNPs@citrate) and silver ions (as a positive control) were tested for their effects on zebrafish embryo development. The toxicity of the silver nanoparticles was lower in this study than in others, but the findings nevertheless demonstrated that they killed and abnormalized zebrafish eggs. The findings of this study have important implications for our understanding of the environmental effects of silver nanoparticles and their potential harmful effects on aquatic life.

## 4. Conclusions

The present research intended to create a green synthesis strategy for the efficient production of AgNPs from an aqueous extract of *T. chebula* dried fruit extract. Analyses relating to ultraviolet (UV) spectroscopy, scanning electron microscopy (SEM), and transmission electron microscopy (TEM) were used to validate the fabricated AgNPs. Antibacterial and antioxidant capabilities, protein leakage study, time-kill kinetics assay, and non-targeted toxicological effects on zebrafish were among the many tests used to assess the *T. chebula*-fruit-mediated AgNPs’ potential biological uses. In conclusion, this study lays the groundwork for future research into the biological uses of AgNPs mediated by *T. chebula* fruit and gives valuable insights into their potential. 

## Figures and Tables

**Figure 1 biomedicines-11-01472-f001:**
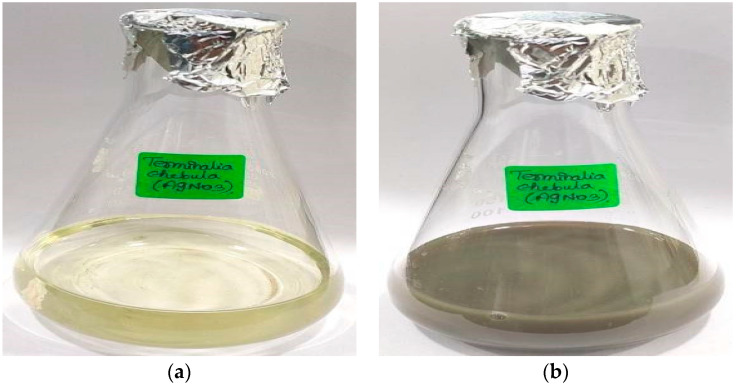
Visual characterization of silver nanoparticles synthesized using *Terminalia chebula*: (**a**) initial color change, (**b**) final color change.

**Figure 2 biomedicines-11-01472-f002:**
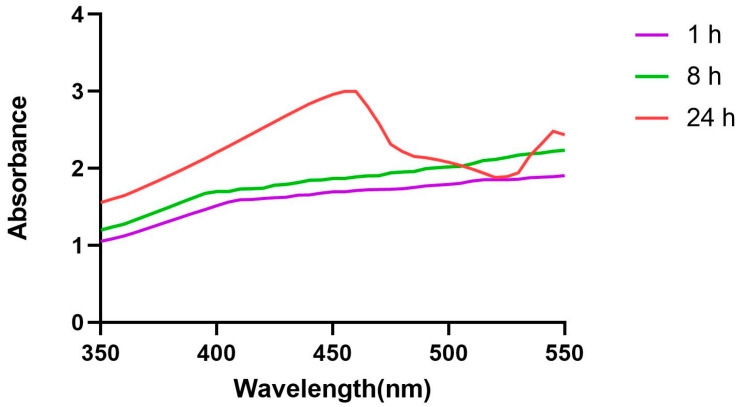
UV-visible absorption spectra of *Terminalia chebula*-mediated silver nanoparticles recorded at various time intervals.

**Figure 3 biomedicines-11-01472-f003:**
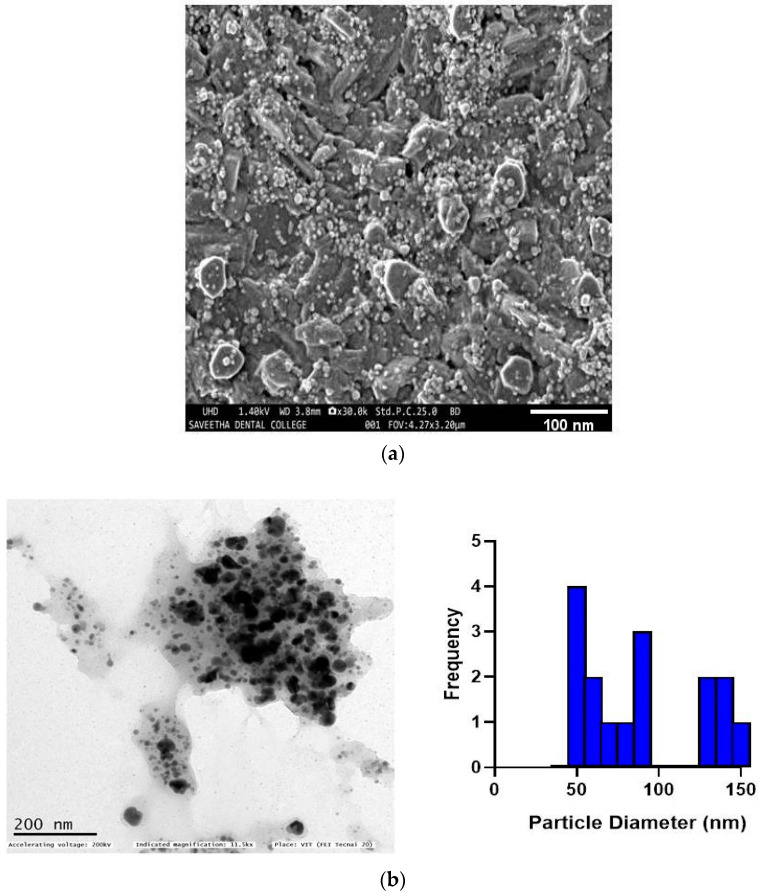
Morphological characterization of silver nanoparticles synthesized using a green approach: (**a**) SEM image, (**b**) TEM image.

**Figure 4 biomedicines-11-01472-f004:**
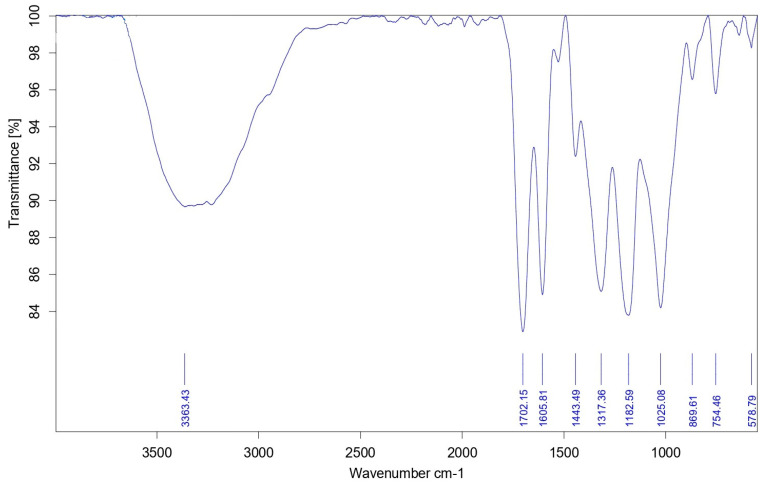
FT-IR spectra of *Terminalia chebula*-mediated silver nanoparticles.

**Figure 5 biomedicines-11-01472-f005:**
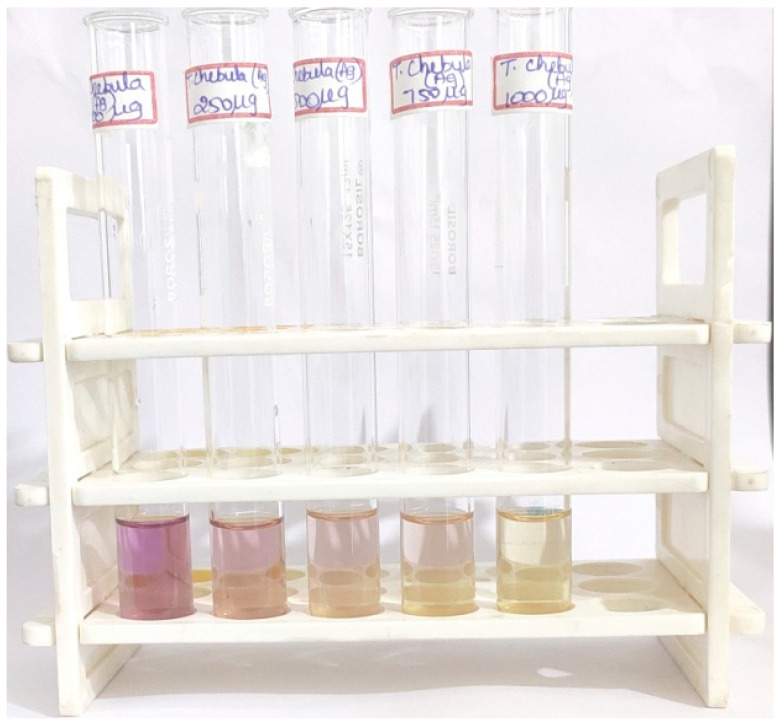
DPPH assay of green-synthesized silver nanoparticles performed at different concentrations.

**Figure 6 biomedicines-11-01472-f006:**
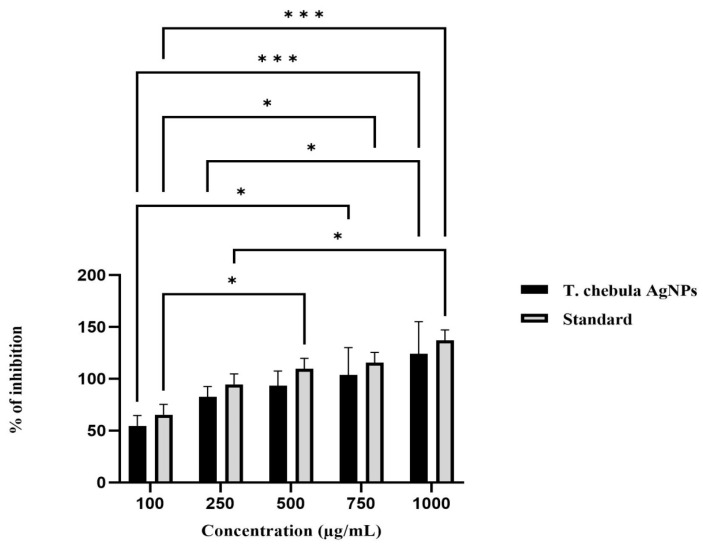
Antioxidant activity of *Terminalia chebula*-mediated silver nanoparticles (TCF-AgNps). The error bar values are expressed as mean ± SE. Significance differences were expressed as * *p* < 0.05, *** *p* < 0.001).

**Figure 7 biomedicines-11-01472-f007:**
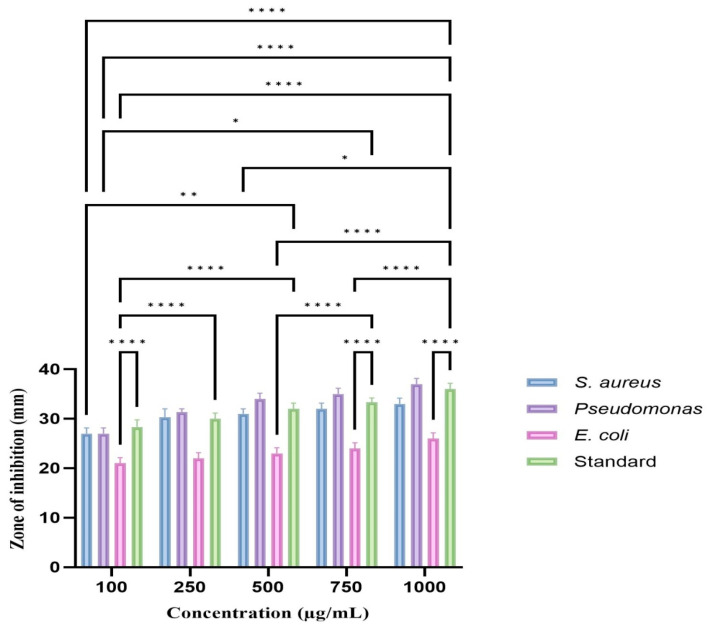
Antibacterial activity of *Terminalia chebula*-mediated AgNPs. The values of the error bars are represented in the format of mean ± standard error. Statistical significance was denoted by asterisks, with * indicating *p* < 0.05, ** indicating *p* < 0.01, and **** indicating *p* < 0.0001.

**Figure 8 biomedicines-11-01472-f008:**
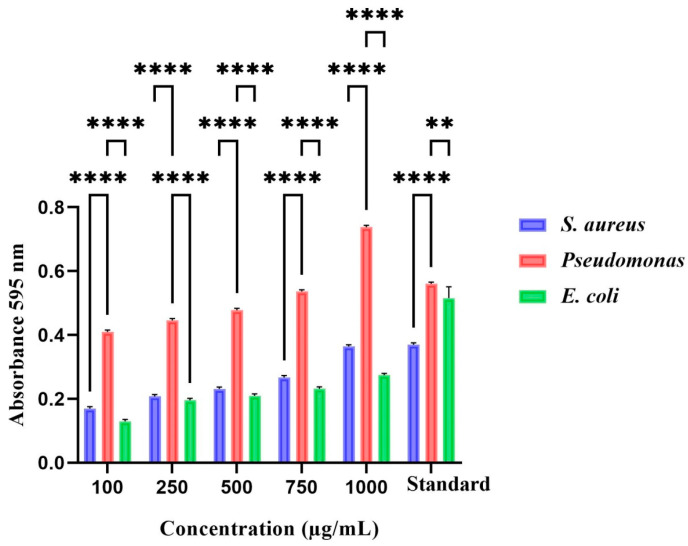
Protein leakage analysis of *Terminalia chebula*-mediated silver nanoparticles against wound pathogens. The values of the error bars are represented in the format of mean ± standard error (SE). Statistically significant differences were denoted as ** (*p* < 0.01) and **** (*p* < 0.0001).

**Figure 9 biomedicines-11-01472-f009:**
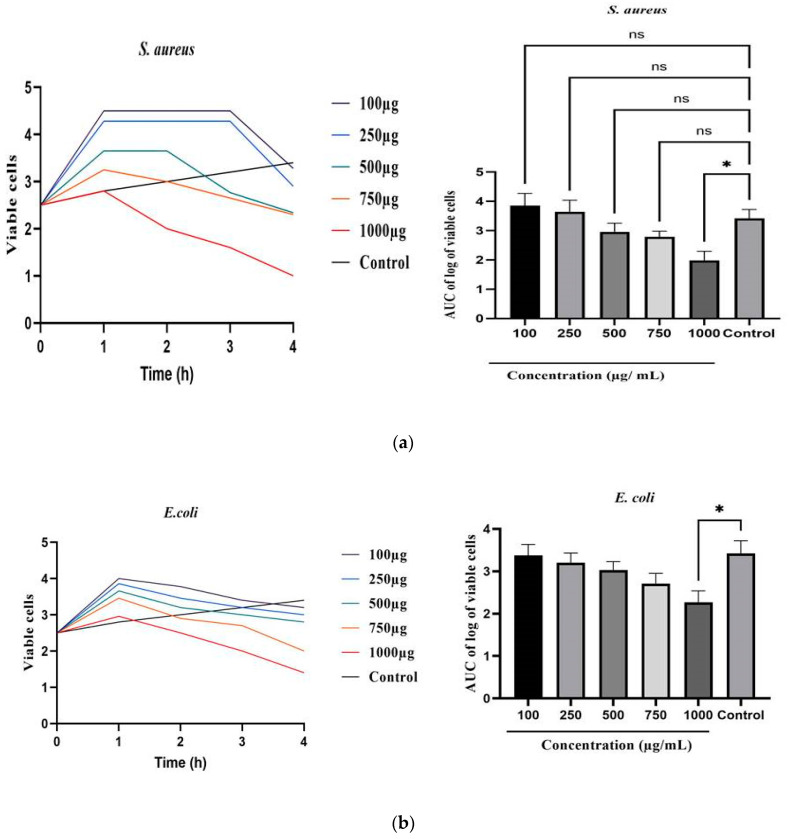
Time-kill kinetics of *Terminalia chebula*-mediated silver nanoparticles; (**a**) *S. aureus*, (**b**) *E. coli.* The values of the error bars are represented in the format of mean ± standard error. Statistical significance was indicated by * *p* < 0.05, while non-significance was denoted by ns.

**Figure 10 biomedicines-11-01472-f010:**
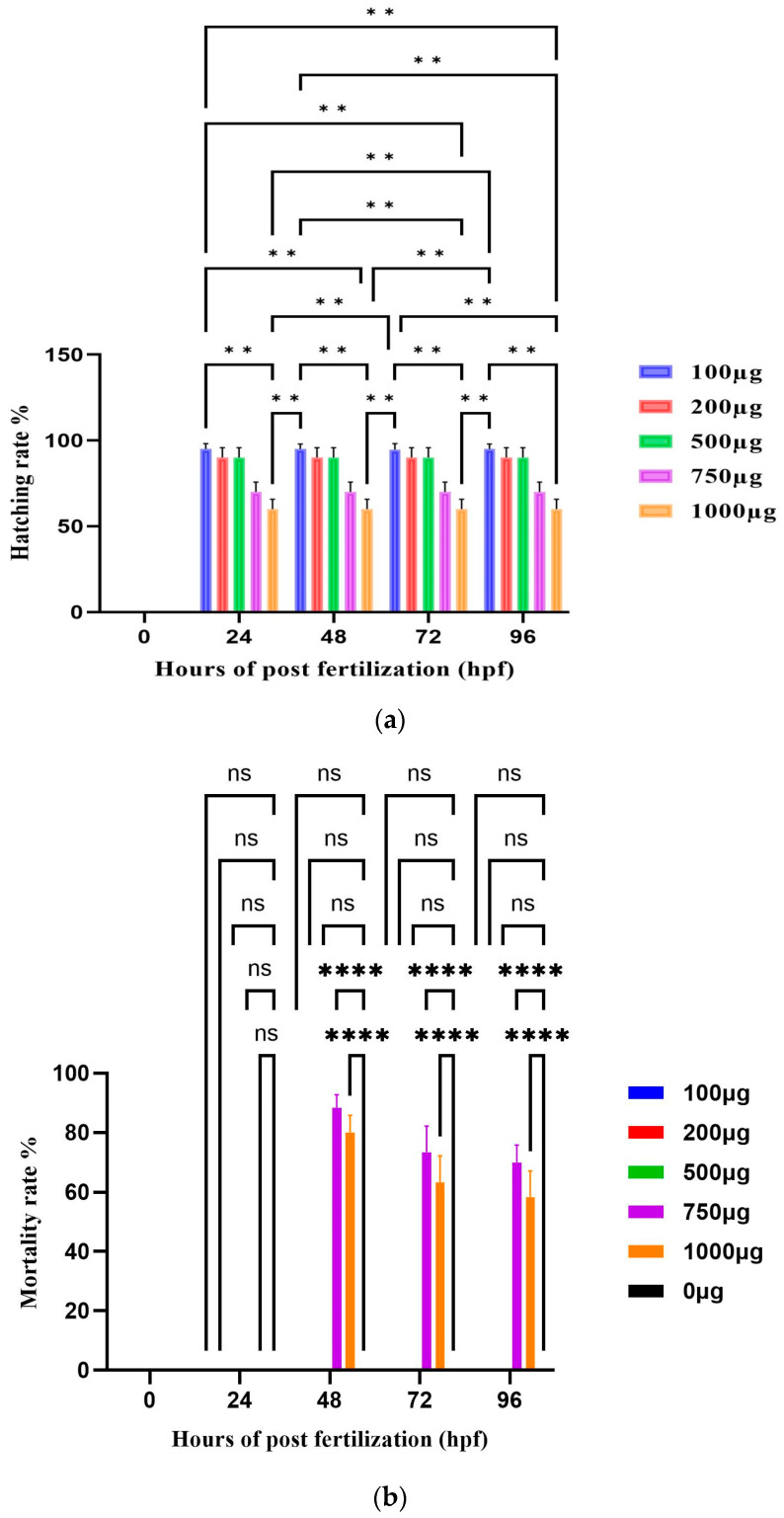
(**a**) The effects of TCF-AgNPs treatment on zebrafish embryo hatchability; (**b**) the impact of TCF-AgNPs treatment on mortality rates in zebrafish embryos. The values of the error bars are represented as the mean value plus or minus the standard error. The statistical analysis revealed that there were significant differences denoted by ** *p* < 0.01, **** *p* < 0.0001, while non-significant differences were denoted as ns.

**Figure 11 biomedicines-11-01472-f011:**
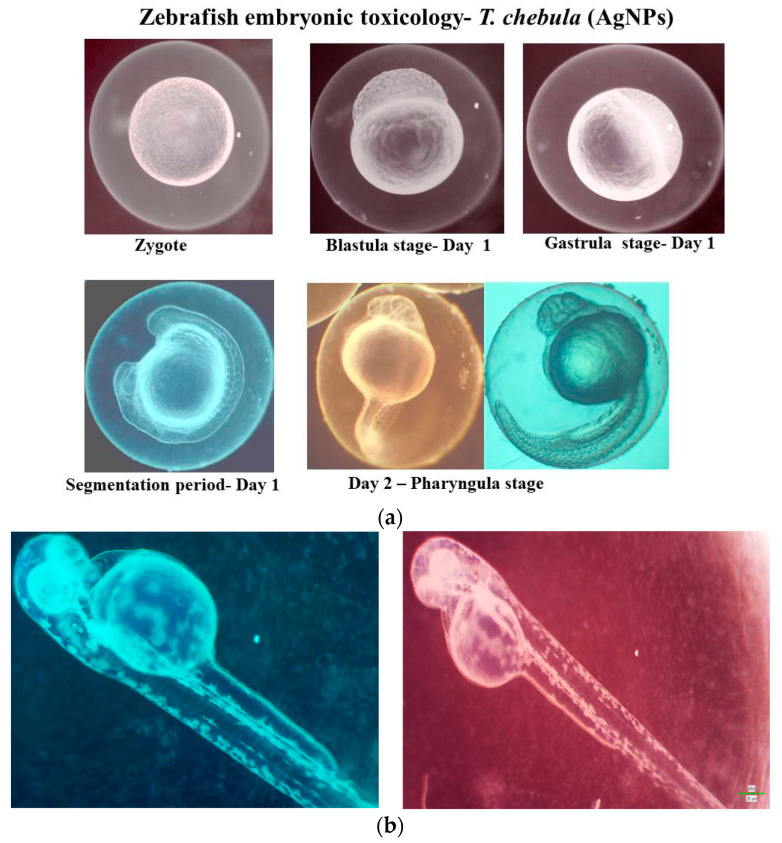
Zebrafish embryonic toxicology evaluation of TCF-AgNPs: (**a**) segmentation period—0–24 hpf, (**b**) 48–72 hpf, (**c**) 96 hpf.

## Data Availability

Not applicable.

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
