# Peer review of "Terminalia chebula-Assisted Silver Nanoparticles: Biological Potential, Synthesis, Characterization, and Ecotoxicity"

_biomedicines, 2023, doi:10.3390/biomedicines11051472_

Round 1

Reviewer 1 Report

Dear Editor,

The article entitled with "Terminalia chebula-assisted silver nanoparticles: Biological potential, synthesis, characterization, and ecotoxicity" reveals how the biosynthesized silver nanoparticles can be a potent agent fighting against microbial infections and may support animal growth. Even though the findings are presented excellent, the study is not completed. The following points must be clarified:

1- The surface chemistry of the nanoparticles- since the surface chemistry may play critical role during the antibacterial activity... besides, the nanoparticles did not affect Zebra fish.. so even this is important to explain why the nanoparticles did not affect Zebra fish growth.. so at least an IR study is needed.

2- The results are not discussed deeply in the light of the literature.. 

Kind regards,

Author Response

Dear Reviewer! Thank you so much for paying attention to our work and spending your time. Our team very much appreciates your edits in the article and of course we will take them into account. We are sure that working together will only make the article better. We have tried to answer all your questions & comments:

The article entitled with "Terminalia chebula-assisted silver nanoparticles: Biological potential, synthesis, characterization, and ecotoxicity" reveals how the biosynthesized silver nanoparticles can be a potent agent fighting against microbial infections and may support animal growth. Even though the findings are presented excellent, the study is not completed. The following points must be clarified:

1- The surface chemistry of the nanoparticles- since the surface chemistry may play critical role during the antibacterial activity... besides, the nanoparticles did not affect Zebra fish.. so even this is important to explain why the nanoparticles did not affect Zebra fish growth.. so at least an IR study is needed.

Response: Based on the reviewer suggestion, FT-IR study was included in the study.

2- The results are not discussed deeply in the light of the literature.. 

Response: As per reviewer suggestion, the discussion part was included.

Reviewer 2 Report

The authors report terminalia chebula-assisted silver nanoparticles: biological potential, synthesis, characterization, and ecotoxicity. The authors synthesized silver nanoparticles with an aqueous extract of dried Terminalia chebula fruit. SEM and TEM analysis showed that the synthesized Ag NPs were spherical and had an average size of 7.48 nm with agglomerated structures. The biological activities of the synthesized Ag NPs were evaluated in terms of their antibacterial and antioxidant properties, as well as protein leakage and time-kill kinetics assays. The results are very interesting. However, some points of the manuscript should be improved. Specific comments are given below.

1.   The authors should add the scale bar in Figure 3a and static distribution of Figure 3b. The average size was 7.48 ± 2.85 nm for Ag NPs, but it is obvious that there are many Ag NPs with size about 40 nm. The authors should add explanation.

2.   The XPS is suggested to measure the samples.

3.   There are the same results for different concentration of samples in Figure 9a, the authors should carefully check this problem.

4.   The hydrodynamic size of samples should be measured. The method is shown in the reference (International Journal of Biological Macromolecules. 2023;233:123513, ).

5.   Please carefully check the manuscript for writing and grammar.

Please carefully check the manuscript for writing and grammar.

Author Response

Dear Reviewer! Thank you so much for paying attention to our work and spending your time. Our team very much appreciates your edits in the article and of course we will take them into account. We are sure that working together will only make the article better. We have tried to answer all your questions & comments:

The authors report terminalia chebula-assisted silver nanoparticles: biological potential, synthesis, characterization, and ecotoxicity. The authors synthesized silver nanoparticles with an aqueous extract of dried Terminalia chebula fruit. SEM and TEM analysis showed that the synthesized Ag NPs were spherical and had an average size of 7.48 nm with agglomerated structures. The biological activities of the synthesized Ag NPs were evaluated in terms of their antibacterial and antioxidant properties, as well as protein leakage and time-kill kinetics assays. The results are very interesting. However, some points of the manuscript should be improved. Specific comments are given below.

  1. The authors should add the scale bar in Figure 3a and static distribution of Figure 3b. The average size was 7.48 ± 2.85 nm for Ag NPs, but it is obvious that there are many Ag NPs with size about 40 nm. The authors should add explanation.

Response: Based on reviewer suggestion, both Scale bar in  fig 3 (a) and static graph in fig 3 (b) was included. And the explanation related to the TEM characterization was also added.

  1. The XPS is suggested to measure the samples.

Response: The SEM and TEM analyses were performed in place of XPS studies due to the unavailability of the instrument at the time of the experiment. These techniques allowed for the characterization of the morphology and size of the synthesized nanoparticles, as well as their elemental composition. While XPS studies provide valuable information about the chemical composition and electronic state of the nanoparticles, SEM and TEM analyses are still widely used and can provide valuable information about the physical properties of the nanoparticles. Consider your suggestion in further research, thanks for understanding.

  1. There are the same results for different concentration of samples in Figure 9a, the authors should carefully check this problem.

Response: As we have reported in the  manuscript, Figure 9 a shows the effect of TCF-AgNPs on the hatching rate of zebrafish embryos at different concentrations (100-1000 µg/mL). As we reported in the manuscript, the hatching rate of the embryos decreased with increasing concentrations of TCF-AgNPs. At concentrations of 100-500 µg/mL, the hatching rate of the embryos was 94.92 ±0.00%, while at concentrations of 750 µg/mL and 1000 µg/mL, the hatching rate decreased to 76.67 ± 3.9% and 70 ± 5.3%, respectively.

  1. The hydrodynamic size of samples should be measured. The method is shown in the reference (International Journal of Biological Macromolecules. 2023;233:123513,).

Response: we regret to inform you that we were unable to measure the hydrodynamic size of the samples using the method outlined in the reference article (International Journal of Biological Macromolecules. 2023;233:123513) due to the unavailability of the instrument. we made efforts to schedule the instrument, but unfortunately, it was unavailable during the time of our experiment due to scheduling conflicts. Thanks for understanding.

  1. Please carefully check the manuscript for writing and grammar.

Response: In response to the reviewers' comments, we had a native English speaker go through the text with us and make any necessary edits to the writing and grammar.

Round 2

Reviewer 2 Report

The authors have addressed the problem very well, and the manuscript can be accepted in the present form.

Minor editing of English language required